# Effect of Fabrication Method on the Thermo Mechanical and Electrical Properties of Graphene Doped PVDF Nanocomposites

**DOI:** 10.3390/nano12132315

**Published:** 2022-07-05

**Authors:** Mamdouh A. Al-Harthi, Manwar Hussain

**Affiliations:** 1Department of Chemical Engineering, King Fahd University of Petroleum and Minerals, Dhahran 31261, Saudi Arabia; 2Department of Materials and Chemical Engineering, Erica Campus, Hanyang University, Ansan 425020-426910, Korea

**Keywords:** dynamic mechanical properties, PVDF, graphene, pressure sensor, nanocomposites

## Abstract

Nanocomposites of poly (vinylidene fluoride) PVDF with graphene nanoflakes (GNF) were prepared using two different routes. Initially, a mix-melting method was used to prepare composites, and their thermal and mechanical properties were evaluated to choose the better method for future experiment and properties investigation. Then, nanocomposite films were prepared by a simple solution-casting technique using a PVDF/graphene solution. In both cases, the amount of graphene was varied to observe and to compare their thermal and mechanical properties. The addition of graphene to the PVDF matrix resulted in changes in the crystallization and melting behaviors as confirmed by DSC analyses. Increasing the graphene content led to improved thermal stability of the PVDF nanocomposites prepared using both methods. Improvements in mechanical properties by the addition of graphene were also observed. Better performance was observed by the nanocomposites prepared by a mix-melting technique suggesting better dispersion and strong interface bonding between PVDF and graphene particles. Thermal and electrical conductivity were measured and compared. Microstructure and morphology were characterized using FTIR, XRD, and SEM analyses.

## 1. Introduction

Among the various polymers, poly(vinyl difluoride) (PVDF) has attracted much attention in various applications because of its excellent stability toward the environment, chemicals, and ultraviolet radiation and its relatively higher temperature and oxidation reactions [1,2]. Moreover, PVDF shows excellent melt-mixing processibility to prepare various composites, sheets, hollow fibers, and thin membranes [1]. PVDF also is soluble in many common solvents, such as dimethyl formamide (DMF) and *N*-methyl-2-pyrrolidone (NMP), which allows fillers to be dispersed in nanocomposites using a solution-casting method. PVDF is a semicrystalline polymer having a polymorph structure with pyroelectric and piezoelectric characteristics [3]. Moreover, it has a high mechanical strength combined with very low creep and relatively high thermal strength of PVDF compared to engineering polymers, such as polystyrene (PS) [2], polyimide (PI) [3], nylon (PA66) [4,5], and polycarbonate (PC) [6]. Additionally, phase nanofiller or fiber-dispersed PVDF nanocomposites exhibited superior results, clearly indicating that PVDF has great potential for the fabrication of various structural- and electrical-sensing devices.

Among the various electro-conductive nanomaterials, graphene has attracted much attention for its high strength, large surface area, and good electrical properties. Thus, graphene-based materials have potential in many applications, such as sensors [7,8], nano-electronic devices, optoelectronic devices, energy storage, and catalysis [4,5,6,7]. Additionally, fabrication of PVDF composites based on second phase dispersed materials (carbon nanostructure, metal, oxides, etc.) for dynamic mechanical-sensing applications has opened a new era of research to obtain sustainable material at lower processing cost with excellent recycling potential and sensing selectivity. The synergetic effect of both components resulted in improved structure and functional properties relative to pure components [9]. Additives to PVDF also improve the thermal and chemical stability and mechanical properties, leading to substantial enhancement in piezo sensitivity and selectivity [10].

Research interest in the development of polymer-based nanocomposites for dynamic- and pressure-sensitive properties has gained tremendous attention to achieve prompt detection with higher sensitivity and selectivity with low cost. Dynamic sensing provides valuable measurements by responding to the electrical behavior of the materials. Piezoelectric polymers, such as PVDF, provide extra value in measuring the dynamic properties of the composites. So far, the piezoelectric polymeric materials are more appropriate candidates for sensor applications due to the piezo electric effect inherent in their molecular structure and orientation [11]. Various nanoparticles, such as silica, Cu, Au, Ag, nanofiller, nanofiber, nanowires, multiwall carbon nanotubes, and graphene with high electrical conductivity, have been used widely as high-performance functional materials with poly(ethylene terephthalates) (PET), poly urethane (PU), and polymethyl siloxane (PMDS) for pressure-sensing applications [12,13,14,15,16,17,18,19,20,21]. However, the use of PVDF as a polymer matrix has not been widely conducted to date.

PVDF composites can be fabricated using various methods. In this paper, melt-mixing and solution-casting methods were used to fabricate PVDF-graphene nanocomposites, and their dynamic mechanical properties were analyzed and discussed.

## 2. Materials

PVDF (Mw~530,000 g/moL) and *N*,*N*-dimethylformamide (DMF) (99.9% pure) were purchased from Sigma Aldrich, St. Louis, MO 63118, USA and were used directly. Graphene (thickness 50–100 nm, diameter 44 ≤ 44 μm, purity 96–99%) was purchased from Grafen Chemical Industries, Turkey, Ankara and was used without further treatment or purification. Figure 1 shows the SEM images of graphene.

## 3. Experimental Section

### 3.1. Fabrication of PVDF-GR Nanocomposites

PVDF composites were prepared using two methods: (1) solution casting, and (2) melt mixing. Figure 2 shows the optical images of samples prepared by two different methods.

### 3.2. Solution Casting PVDF

When PVDF/graphene composites were prepared using a solution-casting method, and different weight percentages of graphene (0.0, 0.5, 2, and 5%) with respect to the polymer were prepared. The required amount of PVDF was dissolved in 50 mL of DMF (*N*,*N*-Dimethyl formamide) at 90 °C under constant stirring for 4 h. Then, this was mixed with a previously prepared stable dispersion of graphene in 40 mL DMF using ultra sonication for 10 min. The stirring was continued for an additional 1 h using a magnetic stirrer @ 400 rpm. The whole mixture was degassed for 10 min in a vacuum oven, then poured into a stainless sheet Petri dish placed on a leveled flat surface and allowed to dry at 50 °C for 3–4 days. The dried films were peeled off carefully.

### 3.3. Melt Blending

PVDF/graphene composites were prepared by a melt-blend method using a Thermo Hakke Minilab II Extruder. Thermo Fisher Scientific, Dieseltr 4, 76227 Karsruhe, Germany, Thermo Fisher Scientific, Dieseltr 4, 76227 Karsruhe, Germany. The required amount of PVDF was added with different weight percentages of graphene (0.0, 0.5, 2.0, and 5.0%) with a counter rotating screw speed of 100 rpm at 190 °C for 10 min. Finally, a test sheet was prepared using a Carver hydraulic press at 165 °C for 12 min. The nomenclature of the samples is explained in Table 1.

## 4. Characterization 

### 4.1. Thermogravimetric Analysis (TGA)

TGA analysis was performed using a TA instruments Thermogravimetric analyzer, SDT Q600. Samples (6 mg) were heated in nitrogen atmosphere with purge flow of 100 mL/min and heating rate of 10 °C/min, from 30 to 850 °C.

### 4.2. Differential Scanning Calorimetry (DSC)

DSC was conducted for the samples at heating and cooling rates of 10 °C/min from −80 to 250 °C under a nitrogen atmosphere using TA instrument DSC Q1000, TA Instruments, 159 Lukens Drive, New Castle, DE 19720, USA. Here, 6 mg samples were again used for analysis. The melting and crystallization behavior were calculated by DSC.

### 4.3. DMA

A Discovery DMA 850 from TA instruments, 159 Lukens Drive, New Castle, DE 19720, USA was used for the analysis. Dynamic mechanical analysis (DMA) and properties of the samples were determined in a temperature range from −80 °C to 130 °C using an oscillation temperature ramp with a constant frequency of 1 Hz. In all cases, rectangular samples were taken from the sheet with average dimensions of 17.5 × 13 × 0.6 mm.

### 4.4. Thermal Conductivity

Thermal conductivities of the composites were measured using a TA instruments Fox 50 Heat flow meter, TA Instruments, 159 Lukens Drive, New Castle, DE 19720, USA. The circular samples were taken from the sheet with diameters of 2 inches.

## 5. Results and Discussion

Dynamic mechanical analysis was performed to determine the various dynamic mechanical properties, such as storage modulus E′, loss modulus E″, and damping coefficient tan *δ* of the prepared samples as a function of temperature from −80 °C. Nanocomposites made using melt mixing generally showed poor dispersion in the matrix and poor interfacial bonding between the polymer and graphene because of the graphene’s high-aspect ratio, very high specific surface area, and high surface energy [18]. In contrast, better dispersion of graphene into the polymer matrix has been reported using the solution-cast method. Thus, dynamic mechanical properties were measured to confirm the dispersibility of graphene and the relative interactions between the polymer and the filler. Figure 3a,b shows the storage modulus of the composites prepared by mix-melting and solution-casting methods, respectively.

Two phenomena were clearly observed. First, the storage moduli of both composites prepared by two different techniques increased with increasing the graphene content. First, 5% dispersed graphene composite exhibited the highest value, followed by 2.0% and 0.5%, respectively. Second, melt-mixed nanocomposites showed much higher storage moduli, i.e., above 30 k MPa, whereas solution-casting nanocomposites exhibited values of only 3–6 k MPa. All samples showed a gradual decrease in storage modulus with increasing the temperature. Storage moduli of the nanocomposites increased with increasing graphene content because of polymer–graphene interactions, a reduction in polymer mobility due to the presence of graphene, increasing viscoelastic properties, and the stiffness of the composites [21]. Graphene nanoplates separate the polymer chain or block the polymer mobility and thus increase the reinforcement during load transfer, which enhances the storage modulus [22,23,24]. Mix-melt fabricated samples are much tougher compared to solution-casting film samples. However, the storage modulus recoverability of the film was much lower. The lower storage modulus of the solution-casting samples might also be due to the presence of solvent and degradation of the PVDF polymer itself. The decrease in storage modulus with increasing temperature was suggested due to energy dissipation involving cooperative motions of the polymer chain [19].

Figure 4a,b shows the loss modulus for various nanocomposites prepared by both methods as a function of temperature (−80 °C to +80 °C). Loss modulus increased with graphene nanoplatelet loading. The glass transition temperatures (T_g_) of both composites were clearly observed. In the case of melt-mix composites, the T_g_ peak at around −40 °C was very sharp for all samples with different loadings of graphene, and loss modulus increased with increasing graphene content from 0.5 wt% to 5 wt%. The loss modulus peak indicates interactions between the filler and matrix. It is also associated with changes in internal frictions between the filler and the matrix, molecular motions, morphology, and the dispersion between the filler and the matrix. Single and strong peaks indicate the good interfacial interactions and no-phase separation. The drop in loss modulus after the maximum peak is suggested by the free movement of the polymer chain present in the system [25]. The beta relaxation of polymer composites at 20 °C and gamma relaxation at 60 °C were clearly observed, which indicates the presence of polymer mobility even after addition of graphene nanoparticles. In contrast, the solution-cast nanocomposites showed a wide T_g_, around −40 °C to −43 °C. The shift in T_g_ with increasing graphene content indicated the stiffness of composites achieved by adding 5% graphene nanoparticles, which reduced the mobility of the polymer chains. The beta relaxation of composites with 0.5% added graphene was clearly observed; however, it was reduced by the further addition of graphene nanoparticles.

Figure 5a,b shows the delta T, which is the main indication of the T_g_ of the composites. The T_g_ of the polymer can be altered by the addition of rigid filler particles. The addition of graphene can alter the T_g_ of composites, which is explained by [26]: (i) decreasing the mobility of polymer chains by the addition of graphene; and (ii) restriction of segmental motion by graphene layers.

The Tg of pure PVDF and their composites by mix-melting method were found at around −30 °C; however, it was above −40 °C and more when fabricated by solution-casting method. The glass transition temperatures of nanocomposites mainly depend on the dispersion of filler particles. The very sharp T_g_ of the melt-mixing nanocomposites indicates better dispersion of graphene particles in the PVDF polymer. Although there is a little higher shift in the T_g_ peak by the addition of graphene particles, a sharp peak was observed in the melt-mix nanocomposites. Dispersibility of graphene was later confirmed and analyzed by SEM analysis. A wide T_g_ was observed for solution-cast samples. The T_g_ became wider after the addition of graphene. At higher graphene content, the aggregation and poor dispersion of graphene was reported with a lower and wider T_g_ transition peak [27,28]. Graphene-poor and graphene-rich regions and inhomogeneous graphene distribution were also found to affect the relaxation behavior [24].

Composites prepared by different methods were characterized by measuring the crystallization and melting properties. Crystallization and melting properties of neat PVDF and graphene-dispersed nanocomposites were studied as a function of graphene content. It is well known that the mechanical properties of semicrystalline polymers depend on their crystallinity and the internal microstructure [29]. Figure 6a,b show the DSC scans of the nanocomposites prepared by two different methods.

The DSC crystallization curves in Figure 6a, b were analyzed to evaluate various thermal properties and crystallinities, as shown in Table 2. Table 2 shows the crystallization (T_c_ onset and T_c_ peaks), crystallization of enthalpy (ΔH_c_), and percentage of crystallization (X_c_) as a function of graphene content.

It is evident from the DSC curves that the addition of graphene particles has marginal effect on peak melting temperature of virgin PVDF polymer. The peak melting temperature of the melt-mix PVDF sample was 158.77 °C. However, it increased to 159.46 by the addition of 0.5% graphene. It reached a maximum at 159.78 °C when 5% graphene was added. Similar behavior was also observed in nanocomposites prepared by the solution-casting technique. The peak melting temperature of the solution-cast film PVDF sample was 156.46 °C. However, it increased to 158.70 °C after the addition of 0.5% graphene. It reached a maximum at 159.27 °C when 5% graphene was added. The degree of crystallinity (X_c_) of virgin PVDF films prepared by melt-mixing and with added graphene also are shown in Table 2. Crystallinity of graphene-dispersed nanocomposites showed significant improvement. The X_c_% of PVDF was about 50%. After the addition of only 0.5% graphene, it increased to 58%, which is about a 15% increase. The addition of 5% graphene showed more than a 20% increase in crystallinity. No changes in the onset temperature were observed for the mix-melting composites.

The degree of crystallinity (X_c_) of virgin PVDF film prepared by solution casting and that of graphene added nanocomposites also are shown in Table 2. The crystallinity of graphene-dispersed nanocomposites showed significant improvement, as the X_c_% of PVDF was about 53.38%. After the addition of only 0.5%, it increased to 57.81%, which is a 12% increase. The addition of 5% graphene showed more than a 20% increase in crystallinity. Overall, there was an increase in crystallinity of the polymer composites (both synthesized by melt blending and solution casting) with an increase in graphene content or crystallinity that remained in very close range. However, particularly for MDFG-0.2, the value is an artifact attributed to possible agglomeration [30,31,32]. The onset temperature of mix-melting composites showed no significant changes. Comparing the crystallinity rate of composites made using both methods, the crystallinity was much higher for melt-mixing method. The reason for insignificant changes in the Tm is because the melting and crystallization temperatures of polymer are dictated by the crystalline chains, the longer the chains the higher the Tm. However, crystallinity of polymer, in the case of DSC analysis, is calculated from the enthalpy of melting/fusion, which can be affected by the presence of nanofiller because nanofiller can restrict the chain movements of crystalline parts and thus increasing the crystallinity [32,33,34,35].

The addition of rigid filler generally increases the thermal stability of the polymer. The improvement of the thermal stability of graphene-based nanocomposites has been reported by a few researchers. The addition of graphene increases the thermal stability of various polymers, such as PMMA, PS, and PVDF [36,37,38,39,40,41,42]. Figure 7a,b shows the TGA graphs of thermal degradation, and Figure 8a,b shows the DTG curves of the samples prepared by two different methods in air.

Analyzing the TGA curves of nanocomposites prepared by the mix-melt method present a good thermal stability for 0.5–5% graphene samples, as no significant mass changes occurred until a temperature of 400 °C. Initial onset degradation started at 429 °C for 5% graphene composites and around 425 °C for virgin PVDF and 0.5–2.0% graphene PVDF nanocomposites. The earlier degradation of 5% dispersed graphene than that of the 0.5–2% graphene nanocomposites might be due to the presence of volatile impurities in graphene molecules. The literature reports that there are competing factors for graphene loading to dictate the thermal stability, i.e., the action of graphene as a physical barrier to increase the thermal resistance and the presence of oxygenated functionalities to decrease the thermal stabilities; trend reversal may happen at higher loadings of graphene [43,44]. The onset and maximum temperature of degradation are shown in Table 1. Figure 7a shows that 0.5% graphene nanocomposites have the highest temperature at 485.31 °C, followed by PVDF nanocomposites with 2 and 5% graphene at 483.08 and 476.57 °C, respectively. The difference in the onset and degradation temperatures of MDF-1 and SDF-1 could be explained by the difference in processing techniques. Melt processing is performed at higher temperature/stress, and it could result in degradation of weak bonds in polymer chains. Consequently, the melt-mixed neat PVDF has a lower onset degradation temperature.

The presence of one single peak in DTG curves suggests the degradation of virgin PVDF, and graphene dispersed nanocomposites occurred at one stage until 476 °C. The residue of the nanocomposites increased with increasing graphene content, except for the 0.5% graphene sample because of char formation. The thermal stability of the graphene dispersed nanocomposites showed no significant improvement, which might be associated with the two-dimensional planar structure of graphene particles and might fall under the concept of nano-confinement, as explained by Chen et al. [45,46]_._

Slightly different TGA curves were observed when samples were prepared using the solution-casting method, as shown in Figure 7b. Initial loss in weight for 5% graphene composite samples was observed at 110–120 °C and then 200 °C. This might be explained by the presence of excess solvent or absorbed solvent due to the larger content of graphene (5%), which escaped as temperature increased [15]. The onset temperature of the composites was around 450 °C, which is similar to that in melt-mix samples. However, the maximum degradation temperature was much lower, as shown in Figure 8b and Table 3. The presence of solvent during composite fabrication and polymer chain network degradation can explain the higher degradation. Kim et al. [47] developed a bubble model to explain the thermal stability of polymer nanocomposites. In this model, thermal decomposition of polymer molecular chains produces more and more volatiles during melting with time and higher temperatures until a critical concentration at which the bubbles begin to nucleate. The presence of solvent increases the bubble nucleation, and thus requires less time for the volatiles to reach a critical concentration. Therefore, the thermal stability of the solution-casting sample was slightly reduced compared to the melt-mix sample. The results confirmed that the sample made by solution casting is more thermally unstable than the melt-mix samples.

The thermal conductivity was investigated to understand the effect of the amount of graphene and the dispersibility, and the results are displayed in Table 4. The literature reports that graphene can increase the thermal conductivity of polymeric materials [48,49,50]. However, in this study, we observed contradictory results for the melt-blended samples, MPVDF and MVDFG. The decrease in the thermal conductivity of the melt-blended samples could be attributed to higher specific surface area of graphene, which results in agglomeration of graphene. Similarly, poor dispersion of graphene mismatches between polymer and graphene are produced, and such interface hinders the heat transfer due to phonon scattering [51,52]. However, in the thermal conductivity of the solution-casted samples, SDFG increased to 0.20 and 0.25 W/m/K with 2% and 5% graphene, respectively. Herein, we are comparing these values with the literature values of the solution-casted PVDF, which was reported to be around 0.196 W/m/K in the range of 20–40 °C [53]. This value is also in close range of our measured value of thermal conductivity of MDF-1.

The increase in thermal conductivity of the SDFG composites is due to better dispersion of graphene in the solution-cast PVDF, and it acts as a bridge between PVDF spherulites, thus enhancing heat transfer from spherulites to spherulites. As PVDF is a semicrystalline polymer, thermal interface resistance exists in the boundary between amorphous regions and semicrystalline regions. The graphene sheets’ (GS) bridging of the spherulites enhances heat flow much more efficiently than the neat PVDF. The small increase in thermal conductivity in the mid-range of the GS concentration was attributed to difficulty transferring thermal energy from sheet to sheet.

According to reports, fillers with platelet shapes show advantages in terms of the morphology because of the large contact area, which allows closer contact, leading to reduced phonon scattering [24,54,55]. According to our findings, the thermal conductivity decreased for mix-melting prepared samples, which is not clearly understood. This might be due to the poor interface network between the graphene and the separate region formation of spherulites inside the composites. The increase in the thermal conductivity of the solution-casting samples with 2 and 5% graphene was significant. These results suggest that the graphene in samples prepared by solution casting is in close contact with each other, creating thermally conductive pathways. An increase in conductive pathways also might be due to exfoliation of graphene flakes during fabrication, which were confirmed by XRD analysis.

Figure 9a,b shows XRD analysis curves of nanocomposites prepared by two different fabrication methods. Clear characteristic peaks of pristine graphene are observed at two theta values of 26.6° and at 54.6°, which correspond to an interlayer distance of 3.34 Å in a hexagonal structure at 200 orientations and the 004-crystal plane, respectively. For the melt-mixing method, virgin PVDF resin crystallizes, showing the distinct characteristic peaks at 9.5°, 17.5°, 18.5°, 20.5°, and 27.5°, as shown in Figure 9a. Peaks at 9.5°, 17.5°, 18.5°, and 20.5° are assigned to *α* phase crystals for 100, 020, 100, and 021 reflections, respectively [56]. However, a clear *β* phase was assigned for virgin PVDF at 27.5°. The presence of various amount of graphene in PVDF was also confirmed in the composites. Higher graphene content showed a higher peak intensity at 28.0°. The intensity of the peaks increased with increasing amounts of graphene [15,55,56,57,58].

Similar XRD patterns were observed in solution-cast nanocomposites with slight variations in peak intensity and angle. D-spacing values are shown in Table 5. X-ray diffraction patterns are a common technique to confirm the interlayer expansion of layered powder and crystalline properties of the composites. No shift in the graphene peak confirmed that graphene was exfoliated. However, a significant change in the peak intensity and width of the peaks was observed for samples prepared using the two methods. Samples prepared by the melt-mix method showed narrow peaks and less intensity between theta angles of 14–21°. Only peaks at 26.95° for the 021 plane showed a higher intensity. In contrast, the sample prepared by the solution-casting method showed wide and high intensity between the angles of 14–21°, with increasing d-spacing compared to the melt-mix sample. Only the peak at 26.95° showed lower intensity. This might be explained by the fact that the addition of graphene using a melt-mix method did not allow polymer to enter the graphene layer and thus increase the d-spacing. However, for the solution-casting method, graphene expanded its d-spacing by the presence of solvent and thus shifted to a lower angle, confirming the increase in the d-spacing [59,60,61,62]. For example, the 5% graphene blend nanocomposite shows peaks at 14.3°, 18.83°, 26.97°, and 39.25°, corresponding to d-spacing values of 4.88, 4.7, 3.3, and 2.29Å. However, shifts in peaks and increases in d-spacing were found for peaks at 14.17°, 18.7°, 26.75°, and 36.6°, with spacing values of 6.15, 4.9, 3.4, and 2.45Å. Peak patterns around 40° also confirmed that the higher crystalline nature of the nanocomposites prepared by solution casting, which were confirmed by the DSC analysis results [63,64]. New peaks observed around 18.7° in solution-cast samples were assigned to the 002 plane, which is absent in the melt-mix sample. Analyzing the above facts, it is suggested that with the increase in lamella thickness and crystallinity of nanocomposites, graphene can act as a laminar reinforcing agent coupled with PVDF polymer interlaced in graphene laminar space, which increased the thermal stability of composites.

Figure 10 shows the FTIR overlay graphs of pure PVDF, pristine graphene, and various amounts of graphene dispersed nanocomposites prepared by two different methods. In general, PVDF exhibits at least four crystalline phases α, β, γ, and δ depending on its preparation and temperature. Thus, the vibrational spectrum of PVDF gives valuable information about the structure of PVDF and can be evaluated by FTIR analysis. Ye et al. [65,66,67,68] investigated vibrational spectra of PVDF extensively. Several characteristic peaks (Figure 10a) of PVDF were observed mainly at 480 cm^−1^ (-CF_2_ wagging) *γ*-phase, 530 cm^−1^ (-CF_2_ bending) *α*-phase, 840 cm^−1^ *β* or *γ*-phase (-CH_2_ rocking). Similar peaks were also observed by various investigators [58,59,60]. Additional peaks above 1000 cm^−1^ are proposed for spontaneous polarization [68,69,70]. Characteristic peaks of pristine graphene were observed at 2978, 701, and 544 cm^−1^. Peak intensities of both nanocomposites increased with increasing graphene content. However, the graphene peak at 2978 shifted to lower values at 2916 and 2848 cm^−1^, which might be due to phase transformation of PVDF or good melt-mixing behavior that hindered the –CH stretching within the graphene structure [27]. The thermal stability of melt-mixed nanocomposites showed better performance due the phase transformation and well mixing of graphene with PVDF polymers, as concluded above.

Figure 11 shows the SEM images of the composites prepared by a solution-cast method. SEM images were taken to understand the dispersion quality of graphene, as well as to explain the physical and mechanical properties. Figure 11a shows the PVDF virgin polymer topography of the casted film showing PVDF crystal formation on the surface. The presence of PVDF spherulite structures and sharp fractures were observed.

Figure 11c shows the composites having 0.5% of graphene. Similar topography images were observed. However, the number of spherulites was higher and the fractured surfaces in Figure 11d were sharper than those in Figure 11b. Figure 11e shows composites having 2% graphene. Spherulites formation increased with increasing graphene, which clearly suggests that graphene acts as a nucleating agent to support spherulites growth [71]. Fracture surfaces in Figure 11f also confirmed the presence of PVDF crystals formation and the flat nature of the fracture surfaces. Figure 11g shows composites having 5% graphene. The amount of graphene significantly increased, which is clearly shown in the layer of graphene surrounded by PVDF polymer matrix.

Figure 12 shows SEM images of the composites prepared by the mix-melting method. Figure 12a shows the SEM images of the virgin PVDF prepared by the melt-mix method. In contrast to SEM images of the sample prepared by the solution-casting method, different surface topographies were observed. Melt mixing and flow of polymers during high temperature mixing was clearly observed. The fracture surface image in Figure 12b supported the melt-flow observations. Figure 12b shows the SEM images of the 0.5% graphene PVDF (MDFG-0.5) composites. In contrast to the SEM images of samples prepared by the solution-casting method, MDFG-0.5 sample demonstrated different surfaces morphology. The melt mixing and flow nature of the polymers during high-temperature mixing was clearly observed as found in the virgin polymer. Figure 12c shows the SEM images of the 2% graphene PVDF composites. Similar images were observed in 0.5% graphene composites. However, melt-flow nature was predominant, which clearly indicates the higher toughening phenomenon, which contributes to the higher mechanical properties.

Figure 12d shows the SEM images of the 5% graphene PVDF composites. Graphene-coated polymer can be observed easily in this sample. Good homogenous dispersion and river-like melt flowing, as shown in Figure 12e, suggests the higher mechanical properties of the composites as we observed in our experiments.

The difference in the morphology of MDF and SDF composites is due to different processing techniques. In the melt blending, the polymer is subjected to shear stress and higher temperature, which affect the crystal growth reflected by the melt-flow observation in the morphology. On the other side, the solution-cast sample showed spherulites formation, because spherulites growth is better in viscous solution [72].

## 6. Conclusions

Highly dispersed graphene PVDF nanocomposites were successfully fabricated using two different methods. Mechanical properties, such as storage modulus and loss modulus of those nanocomposites, were investigated, and the melt-mixed composites showed better performance compared to the solution-cast method. SEM investigation showed the better dispersion of graphene in the melt-mixing method. Furthermore, a DSC study and SEM images suggested that the graphene behaves as a nucleating agent during the PVDF crystallization and results in an increase in crystallization temperature with increasing graphene content. Thermal properties, such as TGA and DMA results, confirmed the above findings and confirmed that the addition of graphene improved the thermal properties by increasing the graphene content.

## Figures and Tables

**Figure 1 nanomaterials-12-02315-f001:**
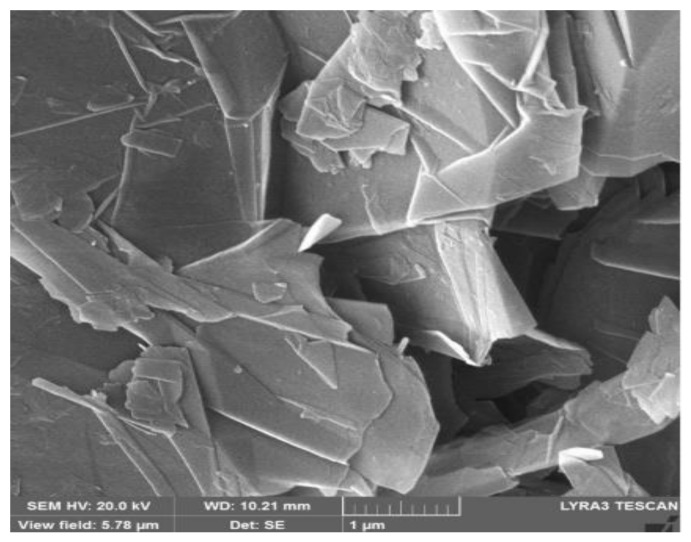
SEM images of supplied graphene.

**Figure 2 nanomaterials-12-02315-f002:**
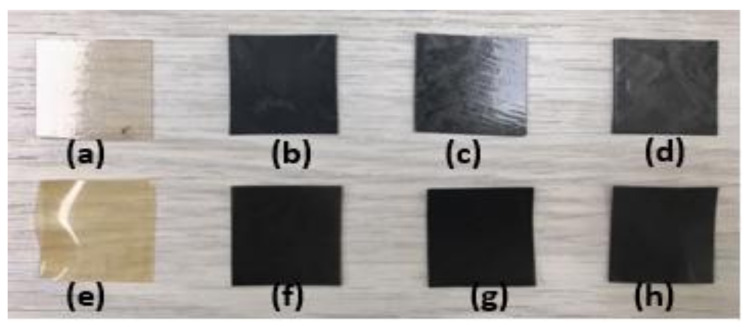
Samples prepared by two different techniques: (**a**) S-DF1; (**b**) S-DFG-0.5; (**c**) S-DFG-2; (**d**) S-DFG-5; (**e**) M-DF-1; (**f**) M-DFG-0.5; (**g**) M-DFG-2; and (**h**) M-DFG-5.

**Figure 3 nanomaterials-12-02315-f003:**
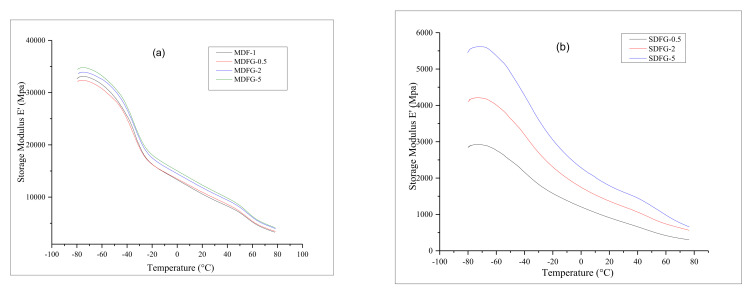
(**a**) Overlay graphs of storage modulus of nanocomposites prepared by melt-mix method and (**b**) prepared by solution-cast method.

**Figure 4 nanomaterials-12-02315-f004:**
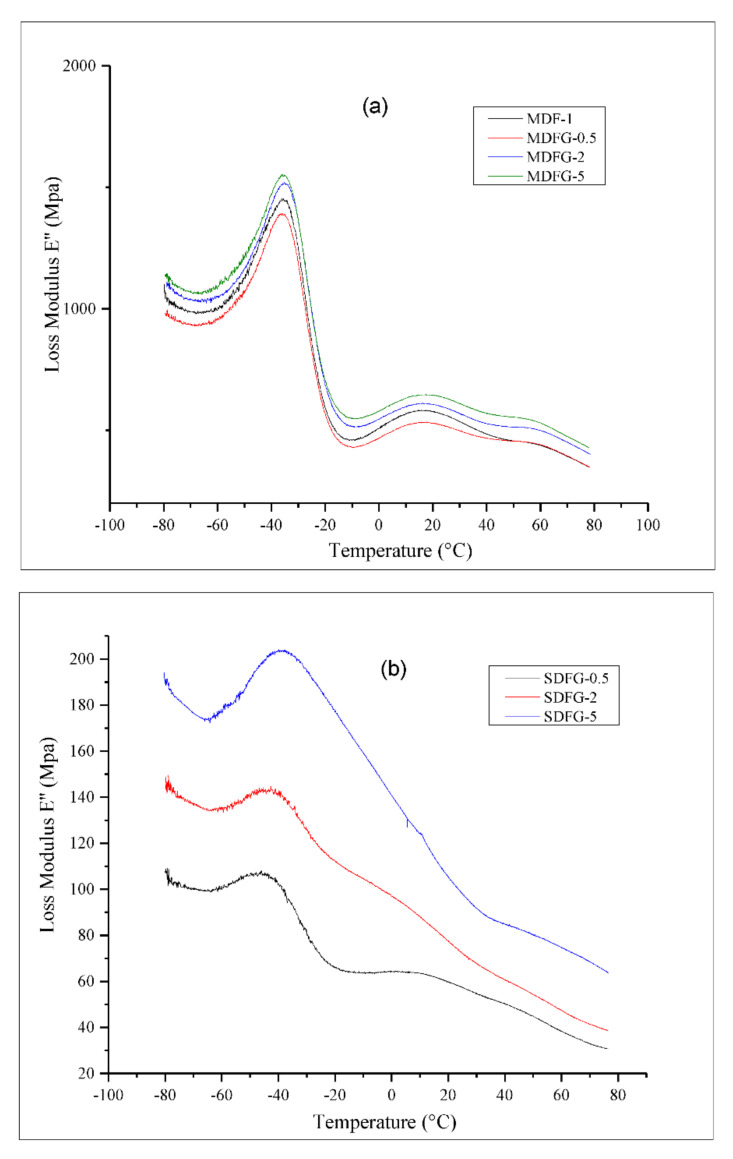
Overlay graphs of loss modulus of nanocomposites: (**a**) prepared by melt-mix method, and (**b**) prepared by solution-cast method.

**Figure 5 nanomaterials-12-02315-f005:**
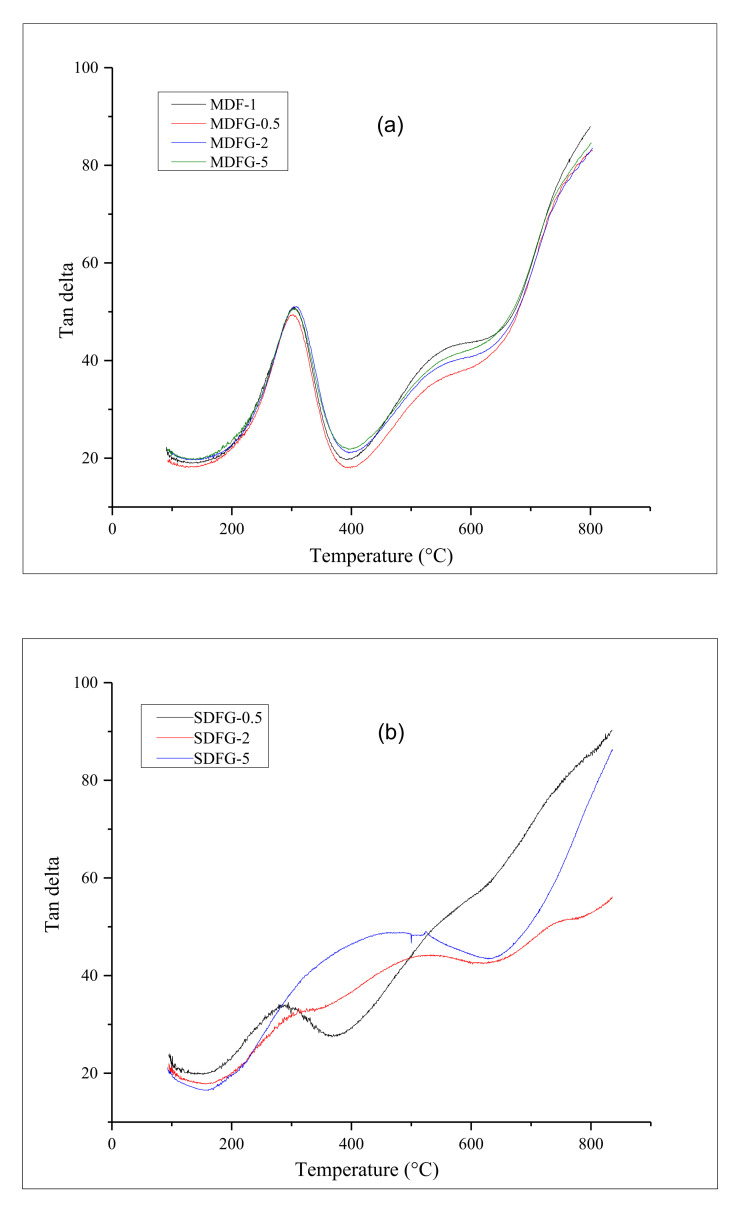
Overlay graphs of tan delta of nanocomposites prepared by: (**a**) melt mix-method, and (**b**) prepared by solution-cast method.

**Figure 6 nanomaterials-12-02315-f006:**
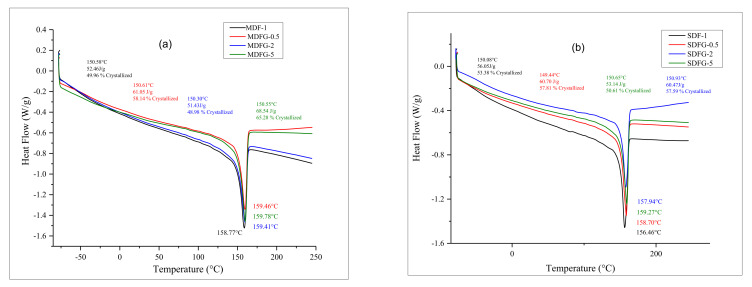
Overlay graphs of DSC of nanocomposites prepared by: (**a**) melt-mix method, and (**b**) solution-cast method.

**Figure 7 nanomaterials-12-02315-f007:**
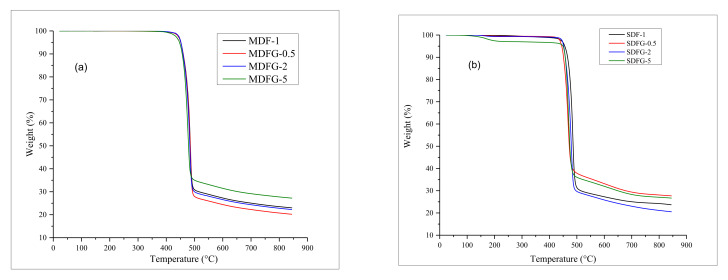
Overlay graphs of TGA of nanocomposites prepared by: (**a**) melt-mix method, and (**b**) solution-cast method.

**Figure 8 nanomaterials-12-02315-f008:**
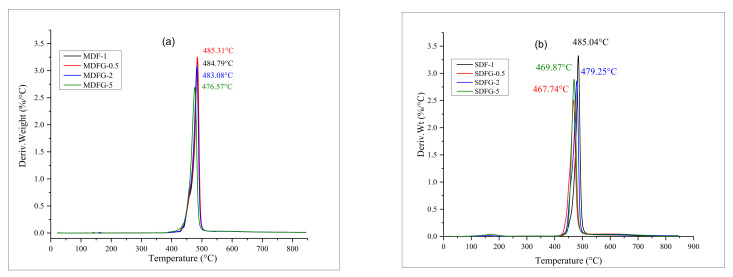
Overlay graphs of DTG of nanocomposites prepared by: (**a**) melt-mix method, and (**b**) by a solution-cast method.

**Figure 9 nanomaterials-12-02315-f009:**
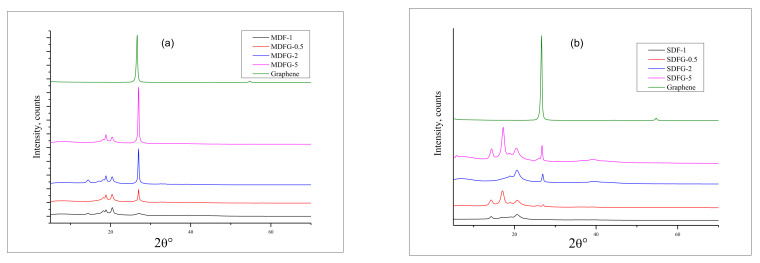
Overlay graphs of XRD of nanocomposites prepared by: (**a**) melt-mix method, and (**b**) the solution-cast method.

**Figure 10 nanomaterials-12-02315-f010:**
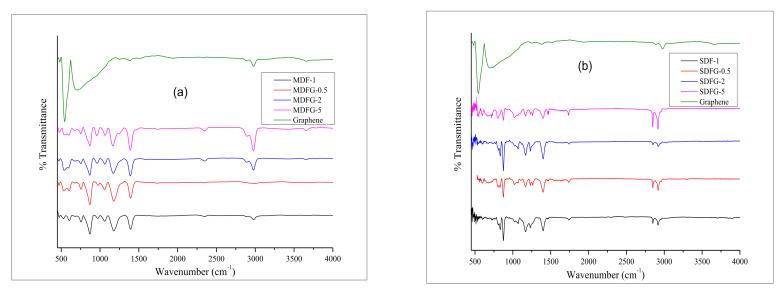
Overlay graphs of FRIT of nanocomposites prepared by: (**a**) melt-mix method, and (**b**) solution-cast method.

**Figure 11 nanomaterials-12-02315-f011:**
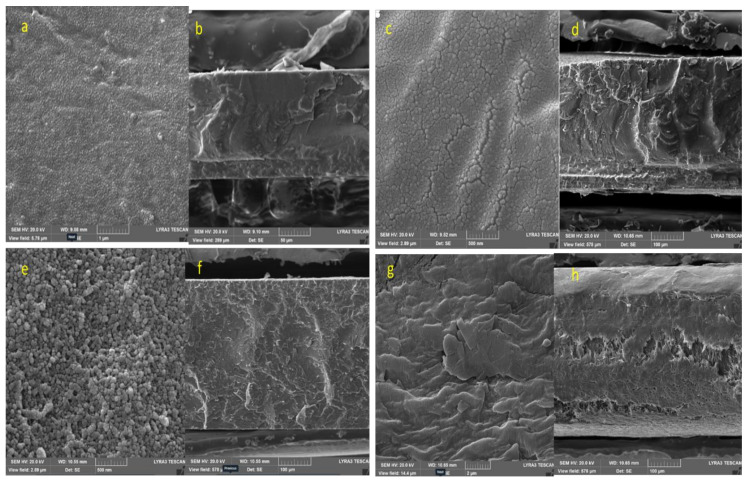
(**a**) PVDF virgin (SDF) (top surface), (**b**) SDF fracture surface, (**c**) SDFG-0.5 (top surface), (**d**) SDFG-0.5 fracture surface, (**e**) SDFG-2 (top surface), (**f**) SDFG-2 fracture surface, (**g**) SDFG-5.0 (top surface), and (**h**) SDFG-5.0 fracture surface.

**Figure 12 nanomaterials-12-02315-f012:**
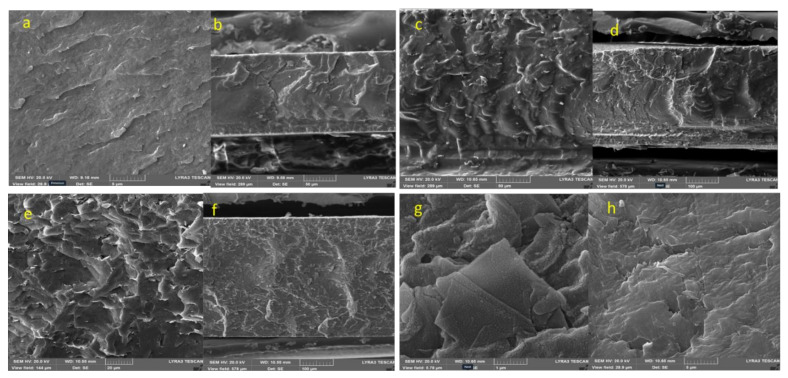
(**a**) PVDF virgin (MDFG0) (top surface), (**b**) MDFG0 fracture surface, (**c**) MDFG 0.5 (top surface), (**d**) MDFG 0.5 fracture surface, (**e**) MDFG 2.0 (top surface), (**f**) MDFG 2.0 fracture surface, (**g**) MDFG 5.0 (top surface), and (**h**) MDFG 5.0 fracture surface.

**Table 1 nanomaterials-12-02315-t001:** Nomenclature of the Prepared Samples.

Sl. No	Sample Name	Composition	Preparation Method	Sl No	Sample Name	Composition	Preparation Method
1	MDF-1	Virgin PVDF	Melt mixing	5	SDF-1	Virgin PVDF	Sol. casting
2	MDFG-0.5	PVDF + 0.5% graphene	Melt mixing	6	SDFG-0.5	PVDF + 0.5% graphene	Sol. casting
3	MDFG-0.2	PVDF + 2% graphene	Melt mixing	7	SDFG-2	PVDF + 2% graphene	Sol. casting
4	MDFG-5	PVDF + 5% graphene	Melt mixing	8	SDFG-5	PVDF + 5% graphene	Sol. casting

**Table 2 nanomaterials-12-02315-t002:** Summary of DSC Results.

Sl. No	Sample Name	T_c onset_ Temp °C	T_c Peak_ Temp °C	X_c_%	H_c_ (J/g)	Sample Name	Onset Temp °C	Peak Temp	X_c_%	ΔH_c_ (J/g)
1	MDF-1	150.38	158.77	49.96	52.46	SDF-1	150.88	156.46	53.38	56.05
2	MDFG-0.5	150.61	159.46	58.14	60.05	SDFG-0.5	149.44	158.70	58.81	60.70
3	MDFG-2.0	150.30	159.78	48.98	51.43	SDFG-2.0	150.05	157.94	60.61	53.14
4	MDFG-5.0	150.55	159.41	68.54	68.54	SDFG-5.0	150.93	159.27	57.59	60.47

**Table 3 nanomaterials-12-02315-t003:** Summary of Thermal Degradation Results.

Sl. No	Sample Name	T_degrad onset_ Temp °C	T_degrad Peak_ Temp °C	Residue wt%	Sample Name	T_degrad onset_ Temp °C	T_degrad Peak_ Temp °C	Residue wt%
1	MDF-1	425	484.79	21	SDF-1S	432	485.04	23
2	MDFG-0.5	423	485.31	20	SDFG-0.5	433	467.74	28
3	MDFG-2.0	427	483.08	22	SDFG-2.0	440	479.25	22
4	MDFG-5.0	429	476.57	30	SDFG-5.0	437	469.87	20

**Table 4 nanomaterials-12-02315-t004:** Thermal Conductivity of the Prepared Samples.

Sl. No	Sample Name	Average Temp °C	Av. Conductivity (W/m∙K)	Sl No	Sample Name	Average Temp °C	Av. Conductivity (W/m∙K)
1	MDF-1	22.5	0.184	5	SDF-1	22.5	-
		32.5	0.184			32.5	-
		42.5	0.183			42.5	-
2	MDFG-0.5	22.7	0.157	6	SDFG-0.5	22.7	-
		32.5	0.157			32.5	-
		42.5	0.157			42.5	-
3	MDFG-0.2	22.5	0.173	7	SDFG-2	22.5	0.202
		32.5	0.173			32.5	0.204
		42.5	0.172			42.5	0.205
4	MDFG-5	22.5	0.157	8	SDFG-5	22.5	0.250
		32.5	0.159			32.5	0.252
		42.5	0.159			42.5	0.252

**Table 5 nanomaterials-12-02315-t005:** d-Spacing Values of Nanocomposites Prepared by Different Methods.

Sl. No	Sample Name	2 θ Angle (Degrees)	Spacing (Å)	Sl. No	Sample Name	2 θ Angle (degrees)	Spacing (Å)
1	MDF-1	14.3 16.9 17.6 20.65 39.38	6.2 5.2 5.6 4.3 2.28	5	SDF-1	14.4 16.9 17.6 20.65 39.38	6.2 5.2 5.0 4.29 2.28
2	MDFG-0.5	18.1 18.82 20.33 26.95 39.23	4.89 4.7 4.36 3.30 2.29	6	SDFG-0.5	17.0 18.8 20.7 25.75 27.02 38.22	5.2 4.6 4.28 3.45 3.29 2.35
3	MDFG-0.2	14.4 18.5 18.84 20.39 26.96 39.26	6.14 4.79 4.7 4.35 3.30 2.29	7	SDFG-2	18.94 20.5 26.9 36.67	4.68 4.32 3.31 2.24
4	MDFG-5	14.3 18.83 20.38 26.97 39.25	4.88 4.7 4.35 3.3 2.29	8	SDFG-5	14.17 18.7 20.5 26.75 36.6	6.15 4.9 4.33 3.4 2.45

## Data Availability

Not applicable.

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
