# Peer review of "Effect of Fabrication Method on the Thermo Mechanical and Electrical Properties of Graphene Doped PVDF Nanocomposites"

_nanomaterials, 2022, doi:10.3390/nano12132315_

Round 1
Reviewer 1 Report
I have read the manuscript provided by the authors and I have to say that it needs a lot of work. The area of graphene with polymers is indeed interesting and has potential, but the manuscript needs to be revised and readjusted.
There are a lot of parts where references must be added to justify the results.
A lot of figures need improvement.
I have a lot of questions that need to be answered.
Please see pdf attached

Author Response
Reviewers reply attached

Reviewer 2 Report
The manuscript entitled “Effect of fabrication method on the thermo-mechanical and conductivity properties of graphene doped PVDF nanocomposites” established two different routes to prepare nanocomposites of poly (vinylidene fluoride) PVDF with graphene nano flakes (GNF). It is a tempting topic in related fields but the paper needs significant improvement before acceptance for publication. The detailed comments are as follows:
1. In this work, the composites are fabricated based on PVDF and GNF. Many similar works using these materials. What are the advantages of this work compared to other similar work using PVDF and GNF? Can you explain it with figure or table?
2. In the manuscript, the author using two different routes to prepare PVDF/GNF nanocomposites. What is the purpose of preparing nanocomposites using different methods? Should more methods be added for comparison?
3. In the manuscript, the author analyzes material properties through characterization. More mechanical-electrical properties of the composites should be tested, such as sensitivity, repeatability and stability.
4. The format of the figures (such as Figure 3 to Figure 12) should be improved. Grammatical errors need to be corrected, please carefully check the errors and modify.
5. These relevant references are suggested IEEE Transactions on Electron Devices, 2019, 66(12): 5407-5410; Carbon, 2022, 187: 35-46.
Round 2
Reviewer 1 Report
I have read the revised manuscript provided by the authors and I have to say it needs more work in order to be considered for publication.
The authors did not respond to all the questions, and also claim to have made the changes or added the references but they did not. There are points that must be clarified before the manuscript can be accepted.
Please see the pdf attached with the comments that were not responded or not corrected by the authors

Author Response
As per the attached file

Round 3
Reviewer 1 Report
Authors have significantly improved their manuscript and now that they have made the appropriate changes and responded to the questions, the manuscript can be accepted.
Author Response
Comment:
In the sentence: "However, the graphene peak at 2978 shifted to lower values at 2916 and 2848 cm-1, which might be due to phase transformation or good melt mixing behavior that hindered the -CH stretching within the graphene molecules [27]..." the last words must be changed, insofar there are not any graphene "molecules".
If the authors mean that the -CH stretching is hindered among the graphene flakes, please tell it explicitly; if they mean that it is hindered in the carbon lattice, then they should mention the graphene lattice. In any case that sentence must be rephrased.
Reply to reviewer.
Thanks for reviewer's comment. I appreciate your cooperation.
Here the author would like to say that the shifting of graphene peak to lower value is due to the phase transformation of PVDF or good melt mixing behavior that hindered the -CH stretching within the graphene structure.
Thus, We have added the word PVDF and changed the word molecules to structure.
